



# Wind Farm Structural Response and Wake Dynamics for an Evolving Stable Boundary Layer: Computational and Experimental Comparisons

Kelsey Shaler[1], Eliot Quon[1], Hristo Ivanov[1], and Jason Jonkman[1]

[1]National Renewable Energy Laboratory, Colorado, USA

**Correspondence:** Jason Jonkman (jason.jonkman@nrel.gov)

**Abstract.** The wind turbine design process requires performing thousands of simulations for a wide range of inflow and control conditions. This necessitates computationally efficient yet time-accurate models, especially when considering wind farm settings. To this end, FAST.Farm is a dynamic wake meandering-based midfidelity engineering tool developed by the National Renewable Energy Laboratory targeted at accurately and efficiently predicting wind turbine power production and structural
loading in wind farm settings, including wake interactions between turbines. This work is an extension to a study into constructing a diurnal cycle evolution based on experimental data. Here, this inflow is used to validate the turbine structural and wake meandering response between experimental data, FAST.Farm simulation results, and high-fidelity large-eddy simulation results from coupled SOWFA-OpenFAST. The validation occurs within the nocturnal stable boundary layer when corresponding meteorological and turbine data were available. To that end, load results from FAST.Farm and SOWFA-OpenFAST are
compared to multi-turbine measurements from a subset of a full-scale wind farm. Computational predictions of blade-root and tower-base bending loads are compared to 10-minute statistics of strain gauge measurements during 3.5 hour of the evolving stable boundary layer, generally with good agreement. This time period coincided with an active wake steering campaign of an upstream turbine, resulting in time-varying yaw positions of all turbines. Wake meandering was also compared between the computational solutions, generally with excellent agreement. Simulations were based on the use of a high-fidelity precursor
constructed from inflow measurements and using state-of-the-art mesoscale-to-microscale coupling.

## 1 Introduction

The wind turbine design process requires performing thousands of simulations for a wide range of inflow and control conditions to capture the structural loads experienced by the turbine over its lifetime. This necessitates computationally efficient yet time-accurate models. When turbines are placed in wind farms, structural loading is also driven by wakes from neighboring turbines





and from wind farm-wide controls strategies, such as wake steering (Fleming et al. (2019, 2020)). To this end, FAST.Farm is a dynamic wake meandering (DWM)-based midfidelity engineering tool developed by the National Renewable Energy Laboratory targeted at accurately and efficiently predicting wind turbine power production and structural loading in wind farm conditions, including farm-wide atmospheric inflows, wake interactions between turbines, and farm-wide control (Jonkman and Shaler (2021)).

Previous FAST.Farm studies show the similarities and differences between FAST.Farm and high-fidelity large-eddy simulations (LES) for rigid and flexible turbines, including wake development and meandering, power performance, and structural loading (Jonkman et al. (2018); Shaler et al. (2019); Shaler and Jonkman (2021)). The first validation of FAST.Farm against measured data took place during the Scaled Wind Farm Technology (SWiFT) benchmark study (Doubrawa et al. (2020)), which showed that underperforming aspects of the simulated wakes were primarily a result of inaccuracies in the inflow and not related to wake modeling itself. But this study did not consider interaction between multiple wind turbines or structural loads. Structural loads calculated by FAST.Farm in single wake conditions (where one turbine is directly upstream of a second turbine) were validated against measurement data from the Alpha Ventus wind farm (Kretschmer et al. (2021)), which showed the importance of wake-added turbulence in low ambient turbulence conditions. In another single wake condition, FAST.Farm was further verified and validated against other engineering models, LES, and measured data from the DanAero wake benchmark study (Asmuth et al. (2022)), which further highlighted the importance of accurate inflow characterization on the turbine response. These validation studies considered two turbines. The only validation of FAST.Farm against measured data with more than two turbines that has been done to date involved the validation of FAST.Farm against five-turbine generator power, rotor speed, and blade pitch results from supervisory control and data acquisition measurements (Shaler et al. (2020)). Despite this verification and validation work, the loads and wake meandering results for multiturbine interactions has yet to be validated.

The objective of this work is to assess the ability of FAST.Farm to accurately predict turbine loads and wake evolution in a small wind farm based on realistic atmospheric conditions, specifically within a nonstationary stable boundary layer. This is done via a three-way comparison between FAST.Farm simulations, high-fidelity LES simulations using the coupled SOWFA-OpenFAST tool, and multiturbine measurements from a small full-scale wind farm, with the simulations driven by a high-fidelity LES precursor using SOWFA of a diurnal cycle derived from measurement-driven mesoscale-to-microscale-coupling (MMC) techniques. The development of the high-fidelity LES precursor is detailed in a companion paper (Quon (2023)). FAST.Farm and SOWFA-OpenFAST simulations are performed for a 3.5-hour nighttime period when atmospheric and turbine data are available, and compared to experimental data from a cluster of five GE 1.5 MW turbines, as shown in Figure 1. This turbine has a hub height of 80 m and rotor diameter of 77 m and a controller supporting variable speed below rated and collective blade pitch-to-feather regulation above rated. The data are collected from the turbines located at the northwest corner of a larger wind farm. 10-minute averages, standard deviations, and power spectra are compared for generator power, rotor speed, blade-root flapwise and edgewise bending moments. Additionally, wake center meandering is compared between FAST.Farm and SOWFA results for all turbines. A portion of the time period studied involved active wake steering of Tr02.

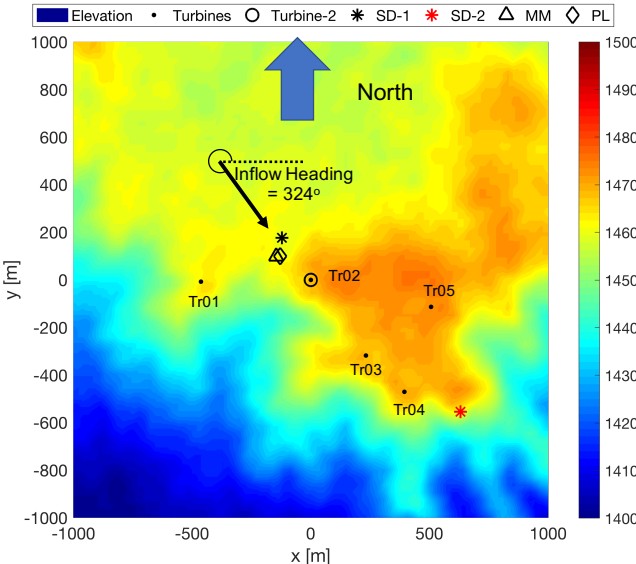

**Figure 1.** Wind farm layout. Tr01—Tr05 indicate the turbine locations. Contours show elevation above sea level in meters. x- and y-axis are easting and northing coordinates, respectively, centered at Tr02. The profiling Doppler lidar (PL) and meteorological mast (MM) are indicated by the diamond and triangle symbols. Sodar (SD) locations are indicated by stars but not used in this work.

## 2 Approach and Methodology

This section provides an overview of FAST.Farm, SOWFA, and experimental measurements, followed by a description of the validation case that was used in this study.

### 2.1 Data Measurements

Data measurements were used to construct the inflow domain used in the FAST.Farm and SOWFA-OpenFAST simulations, and also for validation of the FAST.Farm and SOWFA-OpenFAST structural loads results.

#### 2.1.1 Inflow Conditions

To measure the wind inflow conditions, a 60-m meteorological mast and a WindCube-V2 profiling Doppler lidar are available approximately 160-m upstream of turbine Tr02 along the predominant wind direction. An ultrasonic anemometer on the meteorological mast provides 20-Hz u-, v-, and w-velocity components; additional sensors provide virtual temperature, pressure, and humidity. The profiling Doppler lidar provides 1-Hz wind speed and wind direction data from 40- to 260-m heights with an interval of 20 m. A detailed list of all inflow measurements used to construct the high-fidelity inflow is provided in Quon (2023), which also provides more information on the inflow wind properties, measurements, wake steering campaign details, and why this time period was selected.



### 2.1.2 Turbine Measurements

Two sets of turbine measurements are used for validation in this study. The first contained rotor power measurements from the supervisory control and data acquisition (SCADA) system of all five turbines. This data was collected at 1 Hz and post-processed into 10-minute averages. The second set of measurements contained more comprehensive data from Tr02 and Tr03. These two turbines were instrumented to measure mechanical loads based on guidance from IEC 61400-13, Edition 1. Turbine controller outputs such as rotor power, torque, and speed were provided at 1 Hz and directly integrated with the independent instrumentation into the data acquisition system (DAS). All data was recorded at 50 Hz and stored as 10-minute files. In this study, blade-root and tower-base bending moments were extracted as quantities of interest. The blade-root bending moments were measured $1,500$ mm from the face of the pitch ring. Calibration and scaling was done using a slow rotor roll procedure at two different pitch angles with the blade overhang moment. Tower-base strain gauges were located roughly 6-m above the tower-base flange. A yaw sweep procedure in conjunction with the rotor overhang moment was used to calculate the scale factor. The loads measurement campaign took place from December 10, 2019 through February 16, 2020. However, this work focuses on the 3.5-hour period between $7:30$-$11:00$ UTC on December 26, 2019, as detailed in Quon (2023).

For complete details on the experimental loads campaign, see Ivanov et al. (2021).

## 2.2 Large-Eddy Simulations Setup

High-fidelity LES of the field campaign were performed using SOWFA. This software is based on OpenFOAM version 6 and solves the momentum and potential temperature transport equations for a dry, impressible flow with buoyancy effects represented by the Boussinesq approximation. For the turbine simulations, turbines are represented by actuator disk (AD) and actuator line (AL) models in two distinct simulations. The turbine aerodynamics are loosely coupled to OpenFAST (NREL (2021a)), in which SOWFA passes flow-field velocities to OpenFAST and OpenFAST passes blade forces to SOWFA. (We refer to the coupled software as SOWFA-OpenFAST herein.) The OpenFAST blade forces are represented within SOWFA as a distributed body force, the distribution of which is dictated by a uniform Gaussian kernel with width $\epsilon$. This width is generally chosen to be as small as possible while maintaining numerical stability. For the AD model, the blade forces are distributed with a constant $\epsilon = 3.5$ m and then spread over the entire rotor disk; for the AL model, the blade forces are distributed as a function of blade chord, with $\epsilon/c = 1.6$.

SOWFA simulations were performed in a $4\,\text{km} \times 4\,\text{km} \times 1\,\text{km}$ domain. The precursor simulation was run with uniform spatial discretization of 10 m and temporal discretization of 0.5 seconds. Each of the two simulations with turbines was initiated from the diurnal precursor simulation at $07:30$ UTC on December 26, 2019. In the AD simulation, mesh refinement was added at 2.5 rotor diameters (D) upstream and laterally from all turbines, and extending 15D downstream of Tr04 in the mean wind direction of $337°$. In the refinement region, the spatial discretization was reduced to 5 m and the temporal discretization was reduced to 0.25 seconds. In the AL simulation, the initial refinement was expanded to 10D upstream and laterally 20D downstream. An additional refinement level was added around each turbine that extending 2D upstream and laterally, and 5D downstream. The finest grid spacing was 2.5 m and the temporal discretization was further reduced to 0.1 seconds.



For further details on the SOWFA model and how it was used to generate the inflow, see the companion paper of Quon
(2023).

## 2.3 FAST.Farm Simulations Setup

FAST.Farm is a multiphysics engineering tool that accounts for wake interaction effects on turbine performance and structural loading within wind farms. FAST.Farm is an extension of the NREL software OpenFAST, which solves the aero-hydro-servo-elasto dynamics of individual turbines. FAST.Farm extends this analysis to include wake deficits, advection, deflection, meandering, and merging for wind farms. FAST.Farm is based on the DWM model (Larsen et al. (2008)), but expands on it to address many limitations of past DWM implementations. Using this method, the wake deficit of each turbine is computed using the steady-state thin shear layer approximation of the Navier-Stokes equations and the wake is perturbed with a turbulent freestream to capture wake meandering. Wake merging is modeled using a superposition method (Jonkman and Shaler (2021)).

FAST.Farm simulations were performed using the same precursor generated in SOWFA and used for the SOWFA-OpenFAST simulations. To accomplish this, the SOWFA precursor simulation was sampled at the FAST.Farm low- and high-resolution spatial and temporal sampling frequencies. The high- and low-resolution time steps were at 0.5 seconds and 2 seconds, respectively. High- and low-resolution spatial discretization was 5 m and 10 m, respectively. The low-resolution spatial domain was sized at 2045 km×1100 km×280 km, and the high-resolution spatial domains were sized at $\pm 1.5D \times \pm 1.5D \times 3.6D$, centered around each turbine. Rather than calibrating the wake-related FAST.Farm parameters based on the measured data or SOWFA-OpenFAST results, default values were used.

## 2.4 OpenFAST Model Setup

In the OpenFAST model of each wind turbine, aerodynamic, structural, and controller components were enabled. For FAST.Farm simulations, OpenFAST computes the rotor aerodynamics using the blade-element-momentum (BEM) theory in *AeroDyn15* with advanced corrections, including unsteady aerodynamics. For SOWFA simulations, OpenFAST computes the blade-element part while the induction is accounted for within SOWFA. For all simulations, OpenFAST computes the turbine structural response using *ElastoDyn*, which models the flexibility of the blades, drivetrain, and tower with a combined multi-body and modal structural approach. The controller was modelled using the Reference Open-Source Controller (ROSCO, NREL (2021b)), and is described further in Shaler et al. (2020). A separate controller model was used for Tr02, as described in Shaler et al. (2020). Tower influence on the flow and nacelle blockage, as well as drag on the tower, were not considered.

## 2.5 Validation Cases

Shown in Figure 2 are the time-varying inflow conditions from the nacelle anemometer and the turbine simulations (sampled at hub height, just upstream of the rotor); turbine yaw positions from the nacelle yaw encoder; and the estimated shear exponent from the meteorological mast. The yaw position of each turbine is directly specified through the user-defined controller option in the *ServoDyn* module of OpenFAST, and thus no distinction is made between computational methods and measurements in



**Figure 2.** Time-varying results for measured and simulated inflow velocities at Tr02 (top row) and Tr03 (2rd row), turbine yaw position (3rd row), and ambient shear exponent (bottom). Yaw position and shear exponent results are from measurements. Dots in inflow velocity results show 10-minute averages and bands extend to ±1 standard deviation from the mean.

Figure 2. Yaw positions are centered about a nominal value, such that a yaw position of 0° corresponds to no yaw misalignment





when the inflow wind is primarily at 337°. Time-varying yaw angle settings for Tr02 and Tr03 values were taken directly from experimental data. Due to lack of measurements, Tr04 values were set to be the same as Tr03 values and Tr01 and Tr05 were set to have no yaw misalignment relative to the incoming flow. For more details on these measurements and corresponding uncertainty, see Fleming et al. (2020). The turbine inflow velocities for Tr02 and Tr03 come from experimental measurements and

simulations results. The FAST.Farm and SOWFA-OpenFAST results were taken from the *InflowWind* module of OpenFAST generated at each turbine, computed at the turbine hub location. This simulation output includes wake deficits from upstream turbines, and for SOWFA-OpenFAST only, the induction zone of the turbine whose inflow wind is being output. While there is reasonable agreement between the experimental and FAST.Farm results, the SOWFA-OpenFAST results are consistently lower, especially for Tr03. This is due to the the induction zone upstream of the turbines captured by SOWFA, which reduces

the inflow velocity experienced by the wind turbines. The time-varying shear exponent was computed using profiling lidar data and based on changes to the wind speed between heights of 40 and 120 m. These measurements show a wide range of shear exponents during this time period, and at times large gradients, indicating that the background conditions are not stationary as discussed in Quon (2023). This is important when comparing simulated results and measured data, and this nonstationarity can have a significant impact on the ability of a code to accurately capture rotor response.

2D flow visualizations at hub height of the five turbine simulations are shown in Figure 3. Each row contains results from FAST.Farm (shown as a time average), SOWFA-OpenFAST-ADM (shown as an instantaneous snapshot), and SOWFA-OpenFAST-ALM simulations (shown as an instantaneous snapshot), both without (top) and with (bottom) wake steering. The overall wake trajectory and magnitude is consistent between simulations. These comparisons have been investigated more in previous studies.

In addition to experimental turbine loads comparisons, wake evolution between FAST.Farm and SOWFA-OpenFAST-ALM results are compared. For each turbine, the wake center position was computed using the Simulated and Measured Wake Identification and CHaracterization ToolBox (SAMWICH Box, Quon (2017)), an open-source, Python-based library of wake tracking algorithms. There are several wake-tracking algorithms available in SAMWICH box. The one chosen for this work is the 2D Guassian fit model, which is a function-based wake identification method that solves an optimization problem for the

wake position, two-dimensional shape, and rotation. This method is able to detect the wake center, edge, and shape. This and other wake tracking methods available in SAMWICH Box are discussed in more detail in Quon et al. (2019).

Due to the imperfect nature of the wake tracking algorithm, the resulting wake center time series often includes nonphysical spikes. To minimize this, filtering is required to remove spurious results. For each wake center time series, a median filter was first applied to remove most nonphysical spikes in the data. Any remaining spikes were removed by removing high gradients

in the data, and then a final median filter was applied.

An instantaneous snapshot visualizing the wake center locations is shown in Figure 4. Here, the u-velocity is shown in a plane that is roughly parallel to the unyawed rotor planes, where the black circles represent the projected rotor locations; white circles indicate the region searched by the algorithm to identify the wake center; and a white x shows the calculated wake center, after all filtering has been applied. With visual inspection, these xs appear to be roughly in the center of the wake area

and are indicative that the wake centers are accurately calculated by the algorithm.

(a) FAST.Farm, without wake steering

(b) SOWFA-OpenFAST-ADM, without wake steering

(c) SOWFA-OpenFAST-ALM, without wake steering

(d) FAST.Farm, with wake steering

(e) SOWFA-OpenFAST-ADM, with wake steering

(f) SOWFA-OpenFAST-ALM, with wake steering

**Figure 3.** Time-averaged (FAST.Farm) and instantaneous (SOWFA) 2D flow visualization at 8:30 UTC, sampled at hub height and colored by velocity magnitude normalized by the mean horizontal wind speed.

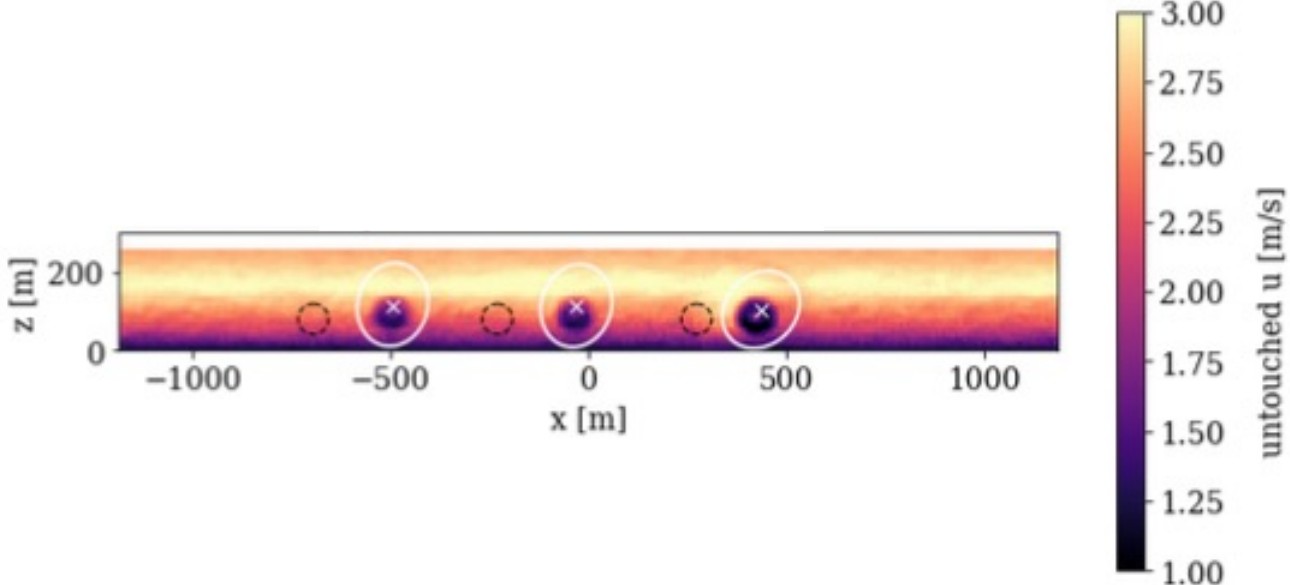

**Figure 4.** Instantaneous snapshot of FAST.Farm u-velocity located 2D downstream of Tr01, 2, and 5. Black circles show projected location of the rotor plane. White circles show the area searched for calculating the wake center of the corresponding turbine wake, with the white x showing the calculated wake center location at 1 hour into the simulation.

## 3 Results

The results of this paper are broken up into two parts. In the first, time-series and power spectral density (PSD) data are compared between experimental measurements and all computational models. In all time series plots, the dots represent 10-minute averages and the shaded regions represent ±1 standard deviation for that 10-minute period. Because this time series data was collected during a wake steering campaign for Tr02, vertical shaded regions are used to show when wake steering of more than ±10° is present (red) and also when there was prominent waking of Tr03 and Tr04 (purple). All PSD plots are focused on key excitation and natural frequencies and do not show the full y-axis range reached (mostly indicative of the mean values whose peak is not shown). In the second part, wake center tracking is used to compare the approximate time-varying wake center position of Tr02, Tr03, and Tr04 for all computational methods.

### 3.1 Turbine Response

Shown in Figure 5 are time-series plots of rotor power for all computational methods and experimental measurements. Here, the experimental data is taken from SCADA measurements; therefore, results are shown for all five turbines, but without the bands for standard deviation that could not be derived from the 10-minute averages. Tr04 has been shown for completeness but excluded from the analysis because, unlike the other turbines, its performance on this day deviated from its operational power curve in both Regions 2 and 3. Additionally, both dimensional (Figure 5a) and non-dimensional (Figure 5b) results are shown.





(a) Dimensional  (b) Non-dimensional

**Figure 5.** Time series results for rotor power for all computational methods and experimental results. Dots show 10-minute averages and bands extend to $\pm 1$ standard deviation from the mean. Results from each wind turbine are shown in separate subfigures. Experimental results for Tr04 are invalid.

For the non-dimensional plots, Tr01 remains dimensional, and the remaining turbines were non-dimensionalized by the corresponding average 10-minute mean value of Tr01 and Tr05 ($\frac{\overline{x}_{Tr01}+\overline{x}_{Tr05}}{2}$). When comparing rotor power of the unwaked turbines (Tr01, Tr02, and Tr05), a primary observation is the that at higher wind speeds FAST.Farm tends to have the highest rotor power for a given 10-minute period, followed by SOWFA-OpenFAST_AD and then SOWFA-OpenFAST_AL. As the wind speed reduces, this order reverses, with SOWFA-OpenFAST_AL results tending to predict the highest power and FAST.Farm predicting the lowest power. There is particularly strong agreement between FAST.Farm and SOWFA-OpenFAST_AD results for Tr01 and Tr05, turbines for which wake steering is never used. When compared to the experimental results, there is overall






strong agreement with the computational results, though some time periods show higher error. In particular, the experimental power of Tr05 is significantly higher than the computation at a few 10-minute periods before 9 : 00 UTC. This is expected to
be caused by an unmeasured spatial variation of the inflow (horizontal gradient). Note that the strongest agreement between computational and experimental results is for Tr02. Though this was the turbine for which wake steering was used, the precise yaw angle was prescribed to match experimental measurements, which is likely the reason for such close agreement, combined with the turbine being unwaked.

When comparing the response of waked turbines (Tr03 and Tr04), discrepancies vary based on how many wakes are impact-
ing the turbine. For Tr03, the same relative trends are seen between the computational methods, with FAST.Farm predicting the highest rotor power at higher wind speed and and SOWFA-OpenFAST_AL predicting the highest rotor power at lower wind speeds. Additionally, there is strong agreement of all models and experimental results for most of the time period, with larger discrepancies at the lowest wind speeds. For Tr04, there are significant discrepancies between all computational models and experimental data for the duration of the time series, and also large discrepancies between the computational models during
the period with strong waking. The differences in computational results during the period with strong waking is likely due to differences in wake breakdown or wake position. In particular, FAST.Farm predicts the lowest rotor power in this region, which is contrary to the unwaked turbine results. This lower rotor power is likely due to a stronger wake that has not broken down as quickly as the wake from the SOWFA-OpenFAST results. At this time, FAST.Farm does not have a wake-added turbulence model or a curled wake model. The lack of a curled wake model may lead to differences in wake shape and deflection, resulting
in more of the wake from Tr03 impacting that of Tr04. Both of these points are investigated in Section 3.2. FAST.Farm is expected to be inaccurate in waked conditions for turbine Tr04 due to its close proximity to Tr03 (223 m), which is less than 3D, due to the near-wake correction used in FAST.Farm that is only implemented to approximate the effect of pressure recovery on the far-wake solution.

Shown in Figures 6(a,b,c) are time-series results for rotor power, torque, and speed for all computational results and ex-
perimental measurements (not SCADA) for Tr02 and Tr03. Results for Tr02 are presented dimensionally, while results for Tr03 are non-dimensionalized by the corresponding 10-minute average of Tr02. For rotor power, these time-series results are very similar to those in Figure 5 but not exact due to different measurement instruments. Note that whereas rotor power and speed may be directly measured, the reported torque has uncertainty associated with the strain gage measurement as well as the estimated gearbox and generator loss factor assumed during calibration. The non-dimensionalization of Tr03 results allow for
a clearer view of the effect of wake interaction, with a strong dip in all quantities during the period of wake interaction, shaded by purple. Comparable results are seen between all methods for all quantities, following the trends described for Figure 5. For all quantities, experimental measurements show higher standard deviations throughout the time series.

Shown in Figures 7(a,b) are time-series results for blade-root flapwise and edgewise bending moments for all computational results and experimental measurements for Tr02 and Tr03. These results show overall strong agreement between all com-
putational results and experimental measurements, both for the means and standard deviations. Relative trends between the computational results are the same as for rotor power, with FAST.Farm predicting the highest loads at the higher wind speeds and SOWFA-OpenFAST_AL predicting the highest loads at the lower wind speeds. For the flapwise bending moment, the





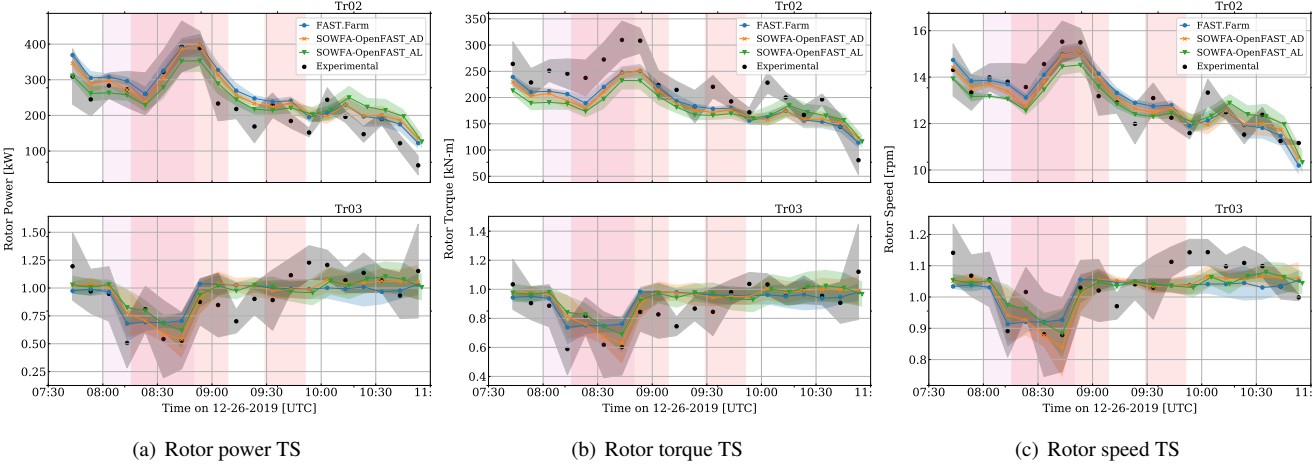

| (a) Rotor power TS | (b) Rotor torque TS | (c) Rotor speed TS |

**Figure 6.** Time series results for rotor power, torque, and speed for all computational methods and experimental results (not SCADA). Dots show 10-minute averages and bands extend to $\pm 1$ standard deviation from the mean. Results from each wind turbine are shown in separate subfigures. Vertical dashed lines indicate the 3P and 6P frequencies based on the average SOWFA-OpenFAST_AD rotor speed.

wake impact on Tr03 is clearly visible, with all normalized results reduced below 1, as well as increased standard deviations, which is generally picked up well by all computational models. The PSD response also compares well for both turbines for

the "good agreement" time period, with clear spikes in all results at the 1P frequency, though the computational results show higher spikes for both turbines. For the "poorer agreement" time period, the results are again comparable, but Tr03 shows much higher spectral content at 1P for the computational results, as well as a spike near the 2P frequency for the SOWFA-OpenFAST results. This is likely caused by higher levels of computed turbulence at this frequency. Edgewise bending moments compare well for all results, both for the time series and PSD results. This is expected as the edgewise bending moment is dominated

by gravity. All results show near-constant means and spectral content peaks at the 1P frequency. The computational results do show higher standard deviations for Tr03, which is likely due to differences in the turbine controllers for Tr02 and Tr03.

Shown in Figures 8(a,b) are time-series results for tower-base fore/aft bending moment for all computational results and experimental measurements for Tr02 and Tr03. Time series computational results for Tr02 compare well to experimental measurements except in the region between $8:30 - 9:00$ UTC where all computational results are nearly $30\%$ lower than

experimental measurements. This time period also corresponds to a region with sharply increasing wind speed, as shown in Figure 2. Relative results for Tr03 compare better in this time period, with the effects of wake interaction captured by all computational methods. When comparing the PSD results, there is overall good agreement for the higher-frequency content, though SOWFA-OpenFAST_AL results tend to have higher spectral content. For experimental measurements, there is a clear spike at $0.2$ Hz, which does not correspond to an $n$P frequency and is not present in any computational results. This spike is

likely due to a rotor imbalance present in the actual turbine that is not captured in the turbine model. Additionally, during the



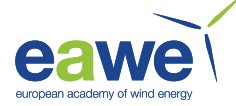


**Figure 7.** Time series (left) and PSD (right) results for blade-root flapwise (top) and edgewise (bottom) bending moments for all computational methods and experimental results. Dots show 10-minute averages and bands extend to ±1 standard deviation from the mean. Results from each wind turbine are shown in separate subfigures. PSD results are shown for two 10-minute time periods; one with good agreement between experimental measurements and computational results (top) and one with poorer agreement (bottom). Vertical dashed lines indicate the 1P, 2P, and 3P frequencies based on the average SOWFA-OpenFAST_AD rotor speed.



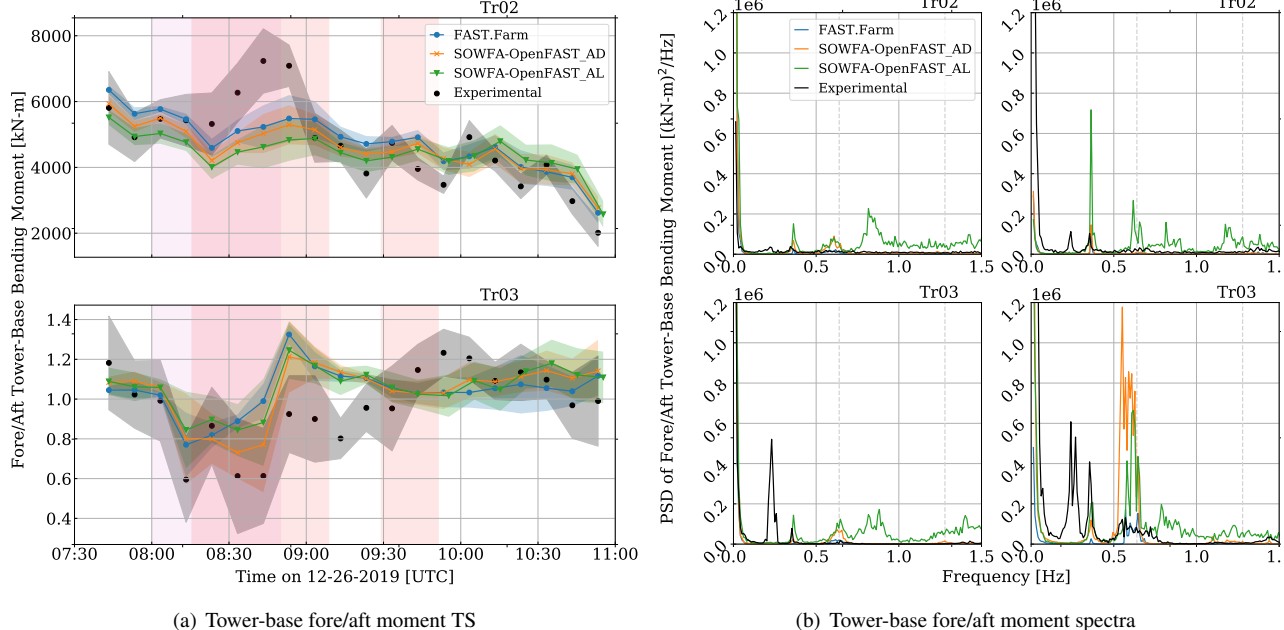

(a) Tower-base fore/aft moment TS       (b) Tower-base fore/aft moment spectra

**Figure 8.** Time series (left) and PSD (right) results of tower-base fore/aft bending moment for all computational methods and experimental results. Dots show 10-minute averages and bands extend to $\pm1$ standard deviation from the mean. Results from each wind turbine are shown in separate subfigures. PSD results are shown for two 10-minute time periods; one with good agreement between experimental measurements and computational results (top) and one with poorer agreement (bottom). Vertical dashed lines indicate the 3P and 6P frequencies based on the average SOWFA-OpenFAST_AD rotor speed

"poorer agreement" time period, the SOWFA-OpenFAST results have much higher spectral content around the 3P frequency which is not present in the FAST.Farm or experimental results.

## 3.2 Wake Center Tracking

Lateral and vertical wake center tracking was performed for all wind turbines and separated into time periods with and without
active wake steering. Shown in Figure 9 are PDF distributions for the lateral and vertical wake center location for each wind turbine at various downstream distances, relative to the wind turbine.

Different wake positions are shown for each turbine based on the availability of information. Recall that Tr01, Tr02, and Tr05 are unwaked turbines, and Tr03 and Tr04 are waked by Tr02 under certain inflow wind directions. When comparing the lateral wake center positions in Figure 9(a), there is comparable agreement between the FAST.Farm and SOWFA-ALM
results at all distances, though FAST.Farm tends to predict more wake deflection at lower downstream distances. A bimodal wake center position is captured for both methods at $9D$ downstream of Tr01, but this could be due to deficiencies in the wake tracking algorithm when wake breakdown occurs. As with Tr01 and Tr05, FAST.Farm tends to predict more wake deflection







**Figure 9.** PDF of lateral (left) and vertical (right) wake center position for all turbines during time periods without wake steering. Results are shown for FAST.Farm and SOWFA-ALM results.

at lower downstream distances, though for Tr02 this persists further downstream. Note that Tr03 and Tr04 are located 5D and 8D downstream of Tr02, respectively. Though SOWFA-ALM results show more wake deflection that FAST.Farm results at 2D

of Tr03, agreement between the computational methods is very good at 5D downstream. Agreement is also very good between the computational methods at 3D downstream of Tr04.



Vertical wake center position results in Figure 9(b) are comparable to those of lateral wake center position in terms of relative difference between computational approaches. The mean wake center positions agrees well between the computational methods for all turbines and downstream locations, though discrepancies in standard deviation are seen more for Tr01 and Tr05 results, especially close to the rotor.

Overall, though, there is strong agreement between the computational methods in lateral and vertical wake center position for all turbines, especially at downstream distances outside of the near wake region, or approximately more than 3D downstream. FAST.Farm is expected to be inaccurate at distances less than 3D downstream due to the near-wake correction used in FAST.Farm that is only implemented to approximate the effect of pressure recovery on the far-wake solution.

# 4 Conclusions

The objective of this work was to assess the ability of FAST.Farm to accurately predict turbine loads and wake evolution in a small wind farm based on realistic atmospheric conditions, specifically a nonstationary stable boundary layer. This was done via a three-way comparison between FAST.Farm simulations, high-fidelity SOWFA-OpenFAST simulations, and multi-turbine measurements from a subset of turbines within a full-scale wind farm, with the simulations driven by a high-fidelity LES precursor of a diurnal cycle derived from measurement-driven MMC techniques. There is generally good agreement between the experimental measurements of turbine response (power, loads) with both computational methods. Overall, there is strong agreement between the computational methods in lateral and vertical wake center position for all turbines, especially at downstream distances outside of the near wake region, or approximately more than 3D downstream. This demonstrates the importance and power of creating highly accurate atmospheric inflow conditions for the use in validation studies.

*Author contributions.* KS led the loads and wakes comparison studies and ran all FAST.Farm simulations. EQ led the inflow generation and SOWFA simulations, and assisted in the experimental data post-processing. HI was involved in the experimental loads campaign and assisted in the experimental data post-processing. JJ supervised the validation effort. KS prepared the article, with support from EQ, HI, and JJ.

*Competing interests.* The authors declare that they have no conflict of interest.

*Disclaimer.* The views expressed in the article do not necessarily represent the views of the DOE or the U.S. Government

*Acknowledgements.* This work was authored by the National Renewable Energy Laboratory, operated by Alliance for Sustainable Energy, LLC, for the U.S. Department of Energy (DOE) under Contract No. DE-AC36-08GO28308. Funding provided by the U.S. Department of Energy Office of Energy Efficiency and Renewable Energy Wind Energy Technologies Office. The views expressed in the article do



not necessarily represent the views of the DOE or the U.S. Government. The U.S. Government retains and the publisher, by accepting the article for publication, acknowledges that the U.S. Government retains a nonexclusive, paid-up, irrevocable, worldwide license to publish or reproduce the published form of this work, or allow others to do so, for U.S. Government purposes.


The research was performed using computational resources sponsored by the Department of Energy's Office of Energy Efficiency and Renewable Energy and located at the National Renewable Energy Laboratory.





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
