# Peer review of "Wind Farm Structural Response and Wake Dynamics for an Evolving Stable Boundary Layer: Computational and Experimental Comparisons"

_Wind Energy Science, 2023_

## Author Comment (AC1)

**Response to Reviewer Comments**

Kelsey Shaler[1], Eliot Quon[1], Hristo Ivanov[1], Jason Jonkman[1]

[1]National Renewable Energy Laboratory, 15013 Denver West Parkway, Golden, CO 80401, USA

*Correspondence to*: Jason Jonkman (jason.jonkman@nrel.gov)

Thank you for supplying us with a thorough review of WES-2023-101. The comments were valuable and we tried to address them all appropriately. In addition to addressing the comments, the document has been reviewed by a professional editor to improve grammar, etc., which resulted in minor editorial changes throughout the paper.

Here are our responses to the specific comments, with the referee comment in green, our response in black, and changes made to the paper indented black.

**Referee #1**

The manuscript discusses a novel comparison FAST.Farm and SOWFA-OpenFAST against measurement data. The work is crucial for the community and very relevant to the readers of Wind Energy Science. I find that the description of the results, their presentation in the figures, and the main findings (what are the main benefits of the different approaches) can be improved; see my list of recommendations below. I hope these suggestions can help to enhance the presentation of the findings.

* Line 154: missing citations;

Author response: Citations have been added for (Jonkman et al. (2018); Doubrawa et al. (2020)).

* Line 162-165: without discussion of the actual parameters, this paragraph is rather vague.

*Author response*: The paragraph in question has been updated and combined with the previous paragraph:

> In addition to experimental turbine load comparisons, the wake evolution between FAST.Farm and SOWFA-OpenFAST-ALM results are compared. For each turbine, the wake center position was computed using the Simulated and Measured Wake tracking algorithms. There are several wake-tracking algorithms available in the SAMWICH ToolBox. The one chosen for this work is the two-dimensional Gaussian fit model, which solves an optimization problem to determine the wake position, two-dimensional shape, and rotation parameters of a Gaussian wake-deficit function. This method is able to estimate the wake center, size, and shape. This and other wake-tracking methods available in SAMWICH Box are discussed in more detail in Quon et al. (2019). Because the wake-tracking algorithm may be sensitive to instantaneous mean wind conditions and the presence of background turbulence structures, the resulting wake center time series can include non-physical discontinuities. To minimize this, filtering is applied to remove spurious results as was done previously by Doubrawa et al. (2020). For each wake center time series, a median filter was first applied to remove the majority of non-physical spikes in the data. Any remaining spikes were removed by eliminating high gradients in the data, and then a final median filter was applied.

* Figure 3: These results are very interesting. It would be great if you could find a way to better demonstrate the differences
between the various models, which, in the current representation, is difficult to judge.

*Author response*: Quantitative comparisons between the models are made later in the paper.

* Figure 4: It is stated that this figure demonstrates that the algorithm captures the wake center location accurately. This is not
so clear from the figure; I would guess the wake centers should be a bit lower. Can you comment on this and how it may
impact the final results?

*Author response*: The following parenthetical comment has been added after this statement in the text to clarify the statement:

(this might not be fully obvious from Figure 4, but is clearer when the wakes are shown with the ambient inflow
subtracted out, which is how SAMWICH processes the wake centers)

* Figure 5: It is unclear why the results for Tr01 are not normalized.

*Author response*: The rationale for showing the non-dimensional results (normalized by the freestream turbines Tr01 and Tr05,
as described in the accompanying text) was to highlight the waked turbine response. As such, we have removed the non-
dimensional subfigures for Tr01, Tr02, and Tr05.

* Section 3.1: The description of figures 5, 6, and 7 is unclear as their explanation is merged, and the reference to the different
figures is unclear.

*Author response*: The results of each figure are discussed in their own paragraphs within this section.

* Figure 6: "Vertical dashed lines indicate the 3P and 6P frequencies based on the average SOWFA-OpenFAST_AD rotor
speed." --> This seems to be a typo.

*Author response*: Changed to:

Vertical shaded regions are used to show when wake steering of more than ±10° is present (red) and when there was
prominent waking of Tr03 and Tr04 (purple).

* Figure 6: Are the lower panels normalized? This is not indicated on the vertical axis

*Author response*: Yes, as indicated in the associated text.

* Figure 6: Define the meaning of the rad bands.

*Author response*: The description has been fixed as indicated above.

* Figure 6: Please define TS. Does this refer to time series?

*Author response*: Yes, TS = Time series. This was previously defined in the caption of Figure 5.

* Figure 7: Make sure text and graphs are not overlapping

* Figure 7: The vertical dashed lines mentioned in the caption are (nearly) invisible. Please make these clearly visible.

*Author response*: We have cleaned up this figure for clarity.

* Figure 7: Define clearer what is defined by good and poorer agreement between model and observations. Looking at the
spectra, the location of the peaks is captured better than in the top panels.

*Author response*: A discussion of this comparison is provided in the text, which explains where the better/worse agreement is
seen.

* Figure 8: Indicate vertical dashed lines.

*Author response*: These are 3P and 6P frequencies as stated in the figure caption.

* Figure 9: Improve alignment of the different panels.

*Author response*: This figure has been cleaned up.

* Line 260: "Though SOWFA-ALM results show more wake deflection that [typo: should be than] FAST.Farm results at 2D
of Tr03, agreement 260 between the computational methods is very good at 5D downstream." --> Can this be discussed in
more detail? [See left middle column]: This result suggests wake development in the different models is different.

*Author response*: Possible reasons for these results have been added to the text.

SOWFA-ALM results show more wake deflection than FAST.Farm results at 2D downstream of Tr03; FAST.Farm
is not expected to accurately model wakes in the near region, but rather, the near-wake model of FAST.Farm exists
so as to more accurately model the far wake. Further downstream of Tr03, agreement between the computational
methods is very good at 5D downstream, as well as 3D downstream of Tr04.

* Conclusion: What is meant by terms like "good" or "strong" agreement should be more clearly defined.

*Author response*: Clarification was made in terms of what showed the agreement. However, a more quantified result (e.g.,
percent difference) is not included due to the nature of the comparisons made in the text.

* Conclusion: I missed a discussion summarizing the benefits and limitations of each approach.

*Author response*: A discussion of SOWFA and FAST.Farm are provided in sections 2.2 and 2.3, respectively. To address this
comment, the following text was added in sections 2.2, 2.3, and 4 respectively:

In general, the AL model requires a finer discretization and is considered higher fidelity than the AD model.

Compared to SOWFA, which resolves the inflow and wakes of the flow field (through the scales resolved by LES),
the flow field in FAST.Farm is solved via engineering models for wave evolution, meandering, and merging atop the
inflow field. The main disadvantage relative to SOWFA is the potentially lower accuracy (hence the need for
validation) and the main advantage being a drastic reduction in computational expense.

Considering that FAST.Farm is much less computationally expensive than SOWFA-OpenFAST, this three-way
validation effort provides further confidence to apply FAST.Farm to the calculation of wind turbine power production
and structural loading in wind farm settings, including wake interactions between turbines.

Typos

Line 201: "and and"

*Author response*: Fixed.

**Referee #2**

Comments on the manuscript entitled "Wind Farm Structural Response and Wake Dynamics for an Evolving Stable Boundary
Layer: Computational and Experimental Comparisons" by Shaler et al. submitted to Wind Energy Science.

In this study, the authors assessed the capability of FAST.Farm in predicting wind turbine loads and wake evolution under
realistic atmospheric conditions by comparing its results with LES and measurements. Evaluating a wind energy model for
real-life conditions is challenging due to the multitude of factors involved. Comments are as follows:

The paper contains vague statements like "good agreement", "excellent agreement", and etc., which require quantifiable
assessments. Moreover, it is not accurate as there are discrepancies as shown in the comparison results. This should be checked
throughout the paper including the abstract the conclusion section.

*Author response*: See our response to a similar comment from Referee # 1.

Regarding Figure 2: If the authors aim to compare the inflows used in FAST.Farm and LES with the measurements, these
should be taken at the same position as the measurements, rather than at the turbine location.

*Author response*: The comparison between measured and LES inflow is included in the companion paper, which focuses on matching conditions at a single location where the profiling lidar and meteorological mast are co-located. Figure 2 shows the inflow conditions extracted from that LES that are directly used in the aero-servo-elastic turbine simulations here.

Accurate inflow is crucial, as emphasized by the authors. Suggestions include adding a brief description of how realistic inflow is generated in FAST.Farm and LES cases, and comparing the time series of inflow wind direction. One more question is raised: Is there a quantitative measure on the accuracy of the employed inflow?

*Author response*: The accuracy of the simulated inflow is discussed at length in the companion paper. An important result that is relevant to this work has been included:

> As discussed in Quon (2023), the mean absolute errors in inflow wind speed, wind direction, and turbulence intensity are 0.19 m/s, 1.5°, and 0.031 (non-dimensional), respectively, during the study period.

Clarify "relative to the wind turbine" on Line 250, Page 14: Is it relative to the averaged wake center or the centerline passing the rotor center in the mean wind direction?

*Author response*: This has been clarified in the text:

> Shown in Figure 9 are probability density function (PDF) distributions for the lateral and vertical wake center location for each wind turbine at various downstream distances, relative to the wind turbine location (e.g., the results for Tr02 are relative to the location of Tr02).

The statement "A bimodal wake center position is captured for both methods at 9D downstream of Tr01, but this could be due to deficiencies in the wake tracking algorithm when wake breakdown occurs." on line 255 page 14: the authors need to clarify whether it is caused by the wake tracking algorithm before drawing conclusions from the figure.

*Author response*: Upon closer inspection, this bimodal response is due to the changing wind direction and resulting change in yaw, which is supported by the yaw misalignment values in Figure 2. The text has been updated to reflect this:

> A bimodal wake center position is captured for both methods at 9D downstream of Tr01. This is due to the changing wind direction and resulting change in turbine yaw misalignment (ranging between +5 and -10 degrees), which has a more pronounced impact on the wake location further downstream of the turbine and is seen developing by 5D downstream of Tr01.

Following from the last comment: does the employed 2D Gaussian fit model work when there are superpositions of wakes?

*Author response*: When the wakes from multiple rotors overlap, SAMWICH does not track the wake of each rotor separately. Rather, the wake center of the "superimposed" wakes is tracked by SAMWICH. While superimposed wakes are most likely not 2D Gaussian in shape, the post-processing with SAMWICH is done consistently across the various results that are compared in this work, and so, the comparison is considered valid.

Typo on Line 160 page 7: "Guassian".

*Author response*: Fixed.

[revised manuscript text omitted]

---

## Author Response (AR2)

**Response to Reviewer Comments**

Kelsey Shaler[1], Eliot Quon[1], Hristo Ivanov[1], Jason Jonkman[1]

[1]National Renewable Energy Laboratory, 15013 Denver West Parkway, Golden, CO 80401, USA

*Correspondence to*: Jason Jonkman ([jason.jonkman@nrel.gov](mailto:jason.jonkman@nrel.gov))

Thank you for supplying us with a thorough review of WES-2023-138. The comments were valuable and we tried to address them all appropriately.

Here are our responses to the specific comments, with the referee comment in green, our response in black, and changes made to the paper indented black.

**Referee #1**

As indicated previously, I believe the work is very important for the wind energy community and should eventually be published. Unfortunately, the main concerns of the referees have not really been addressed in the updated manuscript, and the additional information that is provided in the answers is also limited. Let me rephrase my comments in the hope that this clarifies things. My view remains that it is very important work, but major revision is still required to clarify the presented results.

- Manuscript provides limited details on what exactly has been done and, for many things, refers to previous work. Even in places where clarification was requested, limited information was added, for example, on the wake tracking algorithm (around line 175).

*Author response:* The cited references provide a lot of information that does not need to be repeated here. Nevertheless, we reworded this paragraph with a few more details as follows:

> In addition to experimental turbine load comparisons, the wake evolution between FAST.Farm and SOWFA-OpenFAST-ALM results are compared. For each turbine, the wake center position was computed using the Simulated and Measured Wake Identification and CHaracterization ToolBox (SAMWICH Box, Quon (2017), an open-source, Python-based library of wake-tracking algorithms. There are several wake-tracking algorithms available in the SAMWICH ToolBox. The one chosen for this work is the two-dimensional Gaussian fit model, which solves an optimization problem to determine the wake position, two-dimensional shape, and rotation parameters of a Gaussian wake-deficit function. This method is able to estimate the wake center, size, and shape and was successfully applied to identify wakes under non-neutral atmospheric conditions (Doubrawa et al. (2020)). This and other wake-tracking methods available in SAMWICH Box are discussed in more detail in Quon et al. (2019). Because the wake-tracking algorithm may be sensitive to instantaneous mean wind conditions and the presence of background turbulence structures, the resulting wake center time series can include non-physical discontinuities. To minimize this, filtering is applied to remove spurious results as was done previously by Doubrawa et al. (2020). For each wake-center time series, a moving median filter was first applied to remove the majority of non-physical spikes in the data. A moving median rather than moving mean was applied to help preserve the extrema in the identified wake positions. Any remaining spikes were removed by eliminating high gradients in the position histories that correspond to abrupt changes in wake position, and then a final moving median filter was applied. The resulting filtered trajectories were manually verified to be representative of the simulated wake motion.

- Figures 7 and 8 compare "good" and "poorer" agreements at 10-minute intervals. However, it is unclear how these different intervals are selected. Furthermore, which 10-minute intervals are selected is not mentioned, which would allow comparison to the other graphs.

*Author response:* Regrettably, we no longer have the data to provide the exact time stamps that were used to generated the PSD plots, so, we are left with this qualitative assessment. We have provided some additional clarity in the text on how the time periods were chosen as best as possible.

- I am especially confused about figures 7 and 8. Given the number of panels and the reference to the "top" and "bottom" panels, it is not even exactly clear what panel shows what results. Note, for example, that in Figure 8, all the top panels ("good agreement") are for Tr02, and all the bottom panels are for Tr03. This almost suggests that agreement is "poorer" for Tr03 in general. It is also unclear what the left and right PSD panels indicate. I mentioned this in my previous report [in a different way]. However, unfortunately, the description of what is shown and learned from these figures is still very difficult to understand [changes to these sections are also limited].

*Author response:* Indeed we found some miswording in the captions. This has been corrected for both figure captions.

- Line 154: "This is due to the induction zone upstream of the turbines captured by SOWFA," --> Is this the reason why in Figure 2 "FastFarm" is consistently above SOWFA, while this is not the case in Figure 5. Perhaps it is more how the induction is accounted for? Some clarification on this would be helpful.

*Author response:* Reviewer 1's understanding is correct. The turbine inflow velocity shown in Figure 2 for the SOWFA-OpenFAST results includes the effects of the induction zone upstream of the rotor, while in FAST.Farm, this velocity does not include the induction zone of the rotor, as stated in the text.

- Line 251: "All results show near-constant means and spectral content peaks at the 1P frequency." --> I see significant variations in the peaks at the 1P frequency.

*Author response:* This has been clarified in the text. The authors do believe that there are consistent results between the computational methods for the "good agreement" subplots, which has been better reflected in the updated text.

- Line 257-259: "Relative results for Tr03 compare better in this time period, with the effects of wake interaction captured by all computational methods."

--> Where the results for Tr03 are normalized by the results of Tr02, it really shows something different, namely the wake
effects. The formulation in this paragraph can be clarified to emphasize that point. Differences for Tr02 (reproduction of
atmospheric conditions) and Tr03 due to wake effects.

*Author response:* We have clarified the text to better highlight this point.

- Line 284: "though discrepancies in standard deviation are observed more for Tr01 and Tr05 results, especially close to the
rotor." --> For Tr01 at 9D in Figure 9B, there are also substantial differences, which is not mentioned.

*Author response:* We have now documented the Tr01 differences in the text.

- In general, what is meant by good or poorer agreement remains unclear. Can this be quantified a bit more?

*Author response:* As mentioned in the previous comment, we no longer have the data to provide the exact time stamps that
were used to generated the PSD plots, and so, we are left with this qualitative assessment. We have provided some additional
clarity in the text around Figure 7 on how the time periods were chosen as best as possible.

- Figures general: The text of the horizontal and vertical axis should not overlap.

*Author response:* We rechecked all figures and see that there is negligible text overlap and the figures are easily interpreted
without changes.

- Figure 3: How much wake steering is used in the lower panels? Are there any specific differences the reader should see?

*Author response:* These yaw positions are shown in Figure 2, which has been added to the caption for Figure 3.

- Figure 4: The figure has a very low resolution. As indicated previously, from the figure, it is unclear whether the crosses
correctly reflect the wake center. A not-so-helpful comment is added in lines 183-185. Can you show the corresponding data?
[half of Figure 4 is just white space, so the additional data can be added without making the figure larger]

*Author response:* We have reduced the size of Figure 4, in the hopes that this improves the appearance. As is the issue with
the PSD results, we no longer have access to this data so cannot add these details to the figure. The authors do not see an issue
with the cross placement in the figure, but have also clarified that it is an instantaneous wake center calculation.

- Figure 5: Please clarify in the caption / vertical axis what is plotted in the right column; now, this is only defined in the text.

*Author response:* The caption of Figure 5 has been clarified as follows:
Time-series (TS) results for rotor power of all turbines for all computational methods and experimental results. The
supblots on the left (a) are dimensional and on the right (b) are non-dimensionalized. Dots show 10 minute averages
and bands extend to ±1 standard deviation from the mean. Results from each wind turbine are shown in separate sub-
figures. Experimental results for Tr04 are invalid.

and lower panels, which is incorrect.

*Author response:* The caption of Figure 6 has been clarified as follows:

Time series results for rotor power, torque, and speed for all computational methods and experimental results (not

SCADA). The supblots on the top are dimensional for Tr02 and on the bottom are non-dimensionalized for Tr03.

Dots show 10 minute averages and bands extend to ±1 standard deviation from the mean. Results from each wind turbine are shown in separate sub-figures. Vertical shaded regions are used to show when wake steering of more than

±10° is present (red) and when there was prominent waking of Tr03 (purple).

- Figure 6 and 8: Why mention "prominent waking of Tr03 and Tr04 (purple)." When there is no Tr04 data presented in this
figure, why is this relevant? I think only the time periods in which Tr03 is waked should be shown [as this is the turbine that
is analyzed]. Arguably, these time periods should only be shown in the Tr03 panels (the waked turbine) and not in the Tr02
panels, reflecting that these panels show different information.

*Author response:* The captions of Figures 6-8 have been edited to remove mention of Tr04.